# Rapid Resolution of Non-Effusive Feline Infectious Peritonitis Uveitis with an Oral Adenosine Nucleoside Analogue and Feline Interferon Omega

**DOI:** 10.3390/v12111216

**Published:** 2020-10-27

**Authors:** Diane D. Addie, Johanna Covell-Ritchie, Oswald Jarrett, Mark Fosbery

**Affiliations:** 1Feline Infectious Peritonitis and Coronavirus Website, 64470 Etchebar, Pyrenees, France; 2Independent Researcher, Maidstone ME14 5BF, Kent, UK; johannacovell@hotmail.com; 3School of Veterinary Medicine, University of Glasgow, Garscube Campus, Bearsden Road, Glasgow G61 1QH, UK; o.jarrett@hotmail.co.uk; 4Newnham Court Veterinary Hospital, Bearsted Road, Weavering, Maidstone ME14 5E, Kent, UK; mark@newnhamvets.com

**Keywords:** feline infectious peritonitis, coronavirus, uveitis, Mutian, adenosine nucleoside analogue, feline interferon omega, mesenteric lymph node, alpha-1 acid glycoprotein, symmetric dimethylarginine

## Abstract

This is the first report of a successful treatment of a non-effusive feline infectious peritonitis (FIP) uveitis case using an oral adenosine nucleoside analogue drug and feline interferon omega, and alpha-1 acid glycoprotein (AGP) as an indicator of recovery. A 2-year-old male neutered Norwegian Forest Cat presented with uveitis, keratic precipitates, mesenteric lymphadenopathy and weight loss. The cat was hypergammaglobulinaemic and had a non-regenerative anaemia. Feline coronavirus (FCoV) RNA was detected in a mesenteric lymph node fine-needle aspirate by a reverse-transcriptase polymerase chain reaction—non-effusive FIP was diagnosed. Prednisolone acetate eye drops were administered three times daily for 2 weeks. Oral adenosine nucleoside analogue (Mutian) treatment started. Within 50 days of Mutian treatment, the cat had gained over one kilogram in weight, his globulin level reduced from 77 to 51 g/L and his haematocrit increased from 22 to 35%; his uveitis resolved and his sight improved. Serum AGP level reduced from 3100 to 400 μg/mL (within normal limits). Symmetric dimethylarginine (SDMA) was above normal at 28 μg/dL, reducing to 14 μg/dL on the cessation of treatment; whether the SDMA increase was due to FIP lesions in the kidney or Mutian is unknown. Mutian treatment stopped and low-dose oral recombinant feline interferon omega begun—the cat’s recovery continued.

## 1. Introduction

Feline coronavirus (FCoV) is a highly infectious enteric virus which causes subclinical infection or diarrhoea in the majority of infected cats [1], but potentially lethal monocyte-associated immune-mediated granulomatous vasculitis [2], known as feline infectious peritonitis (FIP), in around 10% of infected cats [1]. FCoV is a positive-sense RNA alpha coronavirus, and a member of the *Coronaviridae* family of the order *Nidovirales* [3].

Although FIP treatment with an injectable nucleoside analogue [4] and cure of enteric FCoV infection with an oral adenosine nucleoside analogue (Mutian, Nantong Biotechnology, China) have previously been reported [5], this case report the first description of the recovery of a cat with a systemic FCoV infection—i.e., FIP—using Mutian.

## 2. Case Report

Skywise was a 2-year-old male neutered Norwegian Forest Cat from a household of five cats; he presented with uveitis in his remaining eye (the right, he lost his left as a kitten) and intermittent diarrhoea. His iris was discoloured and mutton fat keratic precipitates were visible (Figure 1a).

His history included the introduction of two 11-month-old Norwegian Forest Cats (Link and Zelda) into his household three months previously, and the onset of uveitis five days after being given his annual vaccine booster (Leucofeligen, Virbac, France). Skywise had been tested for FCoV antibodies and was found to have a very high titre of over 10,240 (Idexx Laboratories, Wetherby, UK) three weeks prior to presentation due to an in-contact cat (Paddy) being persistently pyrexic and lethargic. Paddy was negative for FeLV p27 antigens and FIV antibodies (Idexx Laboratories, Wetherby, UK), but his FCoV antibody titre was found to be very high (Table 1), therefore all the other cats in the household were FCoV antibody tested in order to establish if they should be isolated from Paddy, but since they were also found to have high FCoV antibody titres (Table 1) segregation was not necessary.

At presentation, the patient’s body weight was 2.89 kg (Figure 2) and body condition score was 2/9. An ultrasound examination confirmed the absence of any abdominal or thoracic effusion, but a mesenteric lymphadenopathy was present. An ultrasound-guided fine-needle aspirate (FNA) of the mesenteric lymph node (MLN) was taken for FCoV reverse-transcriptase polymerase chain reaction (RT-qPCR) (Idexx Laboratories, Wetherby, UK). The test was positive for mutation M1058L and negative for mutation S1060A [6].

Blood results are shown in Table 2. Haematology revealed a mild to moderate non-regenerative anaemia with a haematocrit (Hct) of 22%, a lymphocyte count of 2.07 × 10^9^/L, and eosinopenia below the detection level (reported as 0.0 × 10^9^/L). Blood biochemistry showed hyperglobulinaemia (77.2 g/L), and an albumin to globulin (A:G) ratio of 0.31. Bilirubin was 11 micromol per litre (μmol/L) (0.64 mg/dL). Alpha-1 acid glycoprotein (AGP) was elevated at 3100 μg/mL (Idexx Laboratories, Wetherby, UK) (normal level is ≤500 μg/mL [7]).

In this table, only parameters which were monitored repeatedly are shown; others for which only few (or no) results were available (e.g., aspartate aminotransferase), or which were not relevant to the progression of this case, are omitted. FCoV antibody titre and faecal FCoV RT-qPCR results are shown in Table 1. Day 0 is the day of the cat’s first diagnosis of FIP; Mutian was given between days 6 and 56, after which the cat was treated with 100,000 units of oral feline interferon omega.

Systemic treatment was instigated with prednisolone 5 mg sid for the uveitis (Table 3). The dose was reduced to 2.5 mg sid, then halted after 7 days and replaced with topical prednisolone acetate 1% ophthalmic suspension (Pred Forte^®^, Allergan, Dublin, Ireland), applied q8 h over two weeks. On the 6th day post presentation, treatment started with an oral adenosine nucleoside analogue (Mutian 200, Nantong Biotechnology, Nantong, China) [5] at 8 mg/kg q24 h in divided doses: this was double the normal dose, in an effort to ensure penetration into the eyeball. S-adenosyl-L-methionine supplementation was recommended to support the liver during Mutian treatment. The Mutian dose was reduced to 6 mg/kg on Day 25, but by then the cat’s weight had increased so that he required the same number of capsules per day.

An FCoV RT-qPCR on faeces (Veterinary Diagnostic Services, University of Glasgow, Scotland) [5] was performed to assess the susceptibility of the strain of FCoV infecting the cat to the antiviral being used; the result was low positive at the threshold cycle (C*_T_*) 30 then negative, showing that the virus was susceptible to the anti-viral (Table 1). Three of four housemates were also positive for FCoV RNA from their faeces (Table 1).

The cat’s weight rapidly increased following the onset of Mutian treatment, whereas it had reduced by 30 g (2.89 to 2.86 kg) while on prednisolone (Figure 2). After 40 days of Mutian, the cat had gained one kilogram (Figure 2). The Mutian dose was adjusted during treatment in accordance with the increase in weight.

Blood testing was repeated on the 13th day of Mutian treatment (18 days post diagnosis) (Table 2). Most importantly, the anaemia had reversed, the Hct had increased from 22.0 to 34.4% and bilirubin had fallen from 11 to 6 μmol/L. Globulin decreased to 57 g/L and albumin increased by 2 g/L, so that the albumin:globulin ratio increased from 0.31 to 0.46, which was still low but improving (Figure 3). Clinical examination revealed that his iris colour was returning to normal (Figure 1, Table 3) and there was an improvement in his sight; his guardian reported an increase in activity—playing with the other cats—and that he had tried to catch a bird. His faeces had regained normal consistency and frequency.

The Mutian dose was reduced to 6 mg/kg from the 20th day of treatment onwards and was stopped at 7 weeks for two reasons: AGP levels had returned to normal (400 μg/mL—normal is <500 μg/mL), and because his SDMA had risen to 28 μg/dL (reference range is below 14 μg/dL). Recombinant feline interferon omega (rFeIFN-ω; Virbagen Omega, Virbac, France) started at 100,000 units q24 h *per os*. SDMA reduced to 19 μg/dL on Day 62 (one week after Mutian was stopped) and to 14 μg/dL on Day 90, which was 35 days after Mutian was stopped (Table 2). Urinalysis performed on days 62, 90 and 153 revealed no abnormalities and the specific gravity was 1.050, 1.055 and >1.050, respectively. Weight gain continued after Mutian was discontinued and feline interferon omega was instigated (Figure 1), although at a slower rate (average of 9 g per day rather than 23 g per day).

The in-contact cats were successfully treated with Mutian at 4 mg/kg for five days to stop them shedding coronavirus, in order to prevent re-infection of the recovered cat; the faeces of all five cats were confirmed negative by RT-PCR (Table 1).

## 3. Discussion

The successful treatment of FIP using anti-viral drugs was first reported by Pedersen [4,9]. Cat guardians have been able to source various anti-FCoV drugs, including Mutian, an oral format of adenosine nucleoside analogue, via the internet. However, such drugs are not licensed for veterinary use, which puts attending veterinary surgeons in a dilemma: they cannot prescribe or supply such treatments, but by providing diagnostic and normal supportive treatment, they can support the clients who do decide to utilise these drugs. In particular, providing an accurate diagnosis is useful, preventing the treatment of about 40% of cats who would otherwise have been erroneously misdiagnosed (data not shown). Our view is that it is preferable that if cat guardians propose to use such medications, even though the products are unlicensed, they should do so under proper veterinary guidance. The purpose of this case study is to report what we found useful in monitoring a non-effusive FIP case being treated with Mutian, followed by feline interferon.

Non-effusive FIP is challenging to diagnose because presenting signs are diverse and the list of differential diagnoses is frequently lengthy. Dunbar et al. 2019 [10], reported that detection of FCoV RNA from a mesenteric lymph node (MLN) FNA is 96% specific for a diagnosis of FIP. Unfortunately, the laboratory initially erroneously performed the FCoV RT-PCR test on the Day 2 blood sample, instead of the MLN FNA as requested—predictably it was negative. FCoV RT-PCR on blood samples is not useful in diagnosing FIP because most cats with FIP are not viraemic by the time of clinical presentation [11] and also because around 5% of cats without FIP test positive [12,13] since FCoV-infected cats go through a transient viraemia [12,13,14,15]. FCoV RNA was found in the cat’s MLN FNA and revealed that it was positive for the mutation M1058L, which was further evidence supporting a diagnosis of FIP [6,11]; although, the methionine to leucine substitution at position 1058 in the FCoV spike protein was found in 89% of tissue samples from 14 cats without FIP, suggesting that the mutation is indicative of a systemic spread of FCoV from the intestine, rather than a virus with the potential to cause FIP [16].

Despite the delay by the laboratory in confirming the FIP diagnosis, we instigated Mutian treatment based on history, clinical signs, blood results, and the elevated AGP, which has previously been shown to be highly specific for differentiating FIP from lookalike conditions [17]. There was a sense of urgency to begin treatment since his FIP staging score was 6, which indicates death within two weeks [18]. The cat’s Hct fell rapidly (from 25.7 to 22% in two days), an indicator which has previously been shown to presage imminent death: Tsai et al., (2011) [18] found that Hct levels dramatically decrease starting 2 weeks before death, and bilirubin increases one week before death.

The cat’s history was typical for many FIP cases: two new eleven-month-old pedigree cats had been introduced into the household which were a likely recent source of coronavirus infection; the cat had received a vaccine booster five days before the onset of uveitis. Both the introduction of new cats and vaccination are stressful for a cat—stress has been reported to precipitate FIP in FCoV-infected cats [19] and Riemer et al., (2016) [20] found that vaccination was the suspected stressor in 6.9% of FIP cases. Unfortunately, the cat’s weight was not recorded between the introduction of new cats and vaccination, so it is unknown if he was already losing weight (i.e., subclinically ill with FIP) when the booster was administered, or if weight loss began afterwards. Nevertheless, this case illustrates that vaccinating FCoV-infected cats may pose a risk for FIP development and that optimally vaccination should be conducted after the cat has ceased shedding the virus, which occurs within a few months in the majority of type 1 FCoV infections [21].

The cat’s clinical signs of weight loss, poor appetite, diarrhoea, and uveitis were also congruent with a diagnosis of non-effusive FIP. FIP is a major cause of uveitis in young cats—ophthalmologists from North Carolina Veterinary School diagnosed FIP in 19 of 120 cats (15.8%) with uveitis [22]. Uveitis was reported in 17 of 59 (29%) cats with non-effusive FIP [23].

Although injectable nucleoside analogue treatment has previously been documented in treating FIP [4], to our knowledge oral adenosine nucleoside analogue treatment of FIP has not previously been recorded, although it has been reported in clearing FCoV infection from asymptomatic or diarrhoeic cats [5]. The patient was given twice the normal FIP treatment dose (i.e., 8 mg/kg vs. 4 mg/kg) to potentially increase the concentration of drug within the eye and to eliminate any virus which might be in the brain, since only 20% of absorbed drug crosses the blood–brain barrier (T Xue, personal communication). The response to treatment was rapid, dramatic, and extremely positive. Although we sought to reduce the dose of Mutian as soon as possible in case of possible side effects, in practice, the quantity of medication did not change significantly because the cat gained weight so rapidly. Pedersen et al., (2019) [4] reported that the simplest indicator of response to treatment was weight gain, and this was spectacular in this case, with the cat gaining 1.1 kg during the first 50 days of treatment. In contrast he had lost 30 g of weight during the week on prednisolone, prior to the onset of Mutian treatment.

In two studies, cats with FIP treated with corticosteroids survived a median of 7.5 days [24] and 8 days [25], which is considerably shorter than the 21 day average reported by Tsai et al., 2011 [18] (although in those two studies the potential adverse effect of taking biopsies on survival has to be taken into account, and many deaths were due to secondary bacterial infection due to massive corticosteroid-induced immunosuppression [25]). In addition, the use of prednisolone treatment was previously found to decrease survival time in cats with FIP that were being treated concurrently with polyprenyl immunostimulant [23]. Therefore, in our case, systemic corticosteroids were replaced with topical corticosteroids to treat his uveitis.

We do not know whether it was the Mutian, systemic and topical steroids, or the combination of all treatments which effected the total resolution of the cat’s uveitis. The cat’s guardian reported some return of vision on Day 6 which was the day Mutian began, indicating that it was the steroids which were responsible for his return of sight (Table 3). Legendre et al., (2017) [23] reported the progress of 3 of 17 cats with ocular manifestations of FIP treated with polyprenyl immunostimulant (PI). Their initial signs included anterior uveitis (all three cats), keratic precipitates (one cat), iris discoloration (one cat), and anisocoria (one cat). In two cats, the anterior uveitis was significantly improved or resolved after 2 months on PI treatment with no corticosteroids. In the third cat, the uveitis did not improve; the cat was receiving topical ocular corticosteroids concurrently with PI, and the eyes were enucleated [23]. Whether or not Mutian alone can effect a cure of FIP-related uveitis without concurrent topical corticosteroids remains to be seen. Systemic corticosteroids would appear to be of little or no value in these cases.

No clinical side effects were observed during Mutian treatment, and the liver and kidney biochemical parameters were within normal limits other than plasma symmetric dimethylarginine (SDMA), which was increased (Table 2). SDMA is an early indicator of kidney disease [26,27]. There are three possible explanations for raised SDMA: first, it can be a consequence of FIP lesions in the kidney. Azotemia is more likely in non-effusive than effusive FIP [20], but as far as we are aware, no study has been performed to establish whether SDMA rises in cats with FIP—veterinary pathologists have speculated that a transient FCoV infection of the kidneys could be the cause of the interstitial nephritis seen in many older cats (W Jarrett and S Toth, personal communication). Second, breed predisposition may be a factor—Birman cats have been shown to have raised SDMA levels [28], but no study has been conducted on SDMA levels in Norwegian Forest Cats, and his SDMA reduced to normal levels, suggesting that raised SDMA was not due to his breed. Third, raised SDMA may have been a side effect of Mutian treatment—SDMA reduced within seven days of stopping Mutian and returned to normal within 35 days after the end of Mutian treatment. However, since we had not tested it earlier in the course of the disease, we do not know if it had been raised prior to the onset of Mutian. During treatment, the cat’s creatinine and urea were always within normal limits, although a progressive rise in creatinine was observed (Table 2), which will be closely monitored in future.

Because the recommended duration of administration of GS-441524 was 84 days [4], cat guardians using other antivirals have treated their cats with FIP for the same period of time. The optimal duration of FIP treatment with oral Mutian has not been determined. However, this case illustrates that treatment may be ended after a much shorter course, provided laboratory indicators for FIP have returned to normal. We believe this case to have fully recovered after only 50 days of treatment for the following reasons—his uveitis resolved, he recovered from anaemia, his lymphopenia reversed, there was a reduction in bilirubin, and a sustained reduction in AGP. Nevertheless, we did not wish to leave the cat totally without antiviral cover, so oral low dose rFeIFN-omega was substituted, as previously described [29]. This drug has antiviral and immunomodulatory properties, and prior to the advent of specific antivirals against FCoV, was—with meloxicamour treatment of choice for FIP [30]. The cat continued to gain weight on rFeIFN-omega, although at a slower rate than previously, levelling out at 4.35 kg after Day 100 (Figure 2), and improvements in haematological parameters continued (Table 2, Figure 3).

Relapse of FIP has not been a problem with this—or indeed other—cases whose treatment we have monitored. We attribute our success to, firstly, assessing that the antiviral is effective against the FCoV strain infecting the cat by monitoring the quantity of virus shed in the faeces (by RT-qPCR) to ensure that the viral load reduces in response to the antiviral; secondly, preventing the re-infection of the FIP-recovered cat by identifying other virus shedders in the same household and stopping FCoV shedding using Mutian [5]; thirdly, to a brief period at a dose which ensures the elimination of virus from the brain; fourthly, to following up nucleoside analogue (or protease inhibitor) treatment with low-dose oral feline interferon treatment until such time as the FCoV antibody titre reduces—an indicator of there being no coronavirus remaining in the body.

Skywise and his housemates are alive and well 6 months post diagnosis.

## 4. Conclusions

To our knowledge, this is the first published case of an oral adenosine nucleoside analogue followed by recombinant feline interferon being used to cure a cat of non-effusive FIP, and of AGP being used as a marker of recovery from FIP. We recommend SAMe supplementation be given along with Mutian to protect the liver, and monitor liver enzyme levels. SDMA levels in future cases following this FIP treatment protocol require further investigation. We suggest that systemic corticosteroids are contraindicated in FIP treatment other than for palliative purposes.

## Figures and Tables

**Figure 1 viruses-12-01216-f001:**
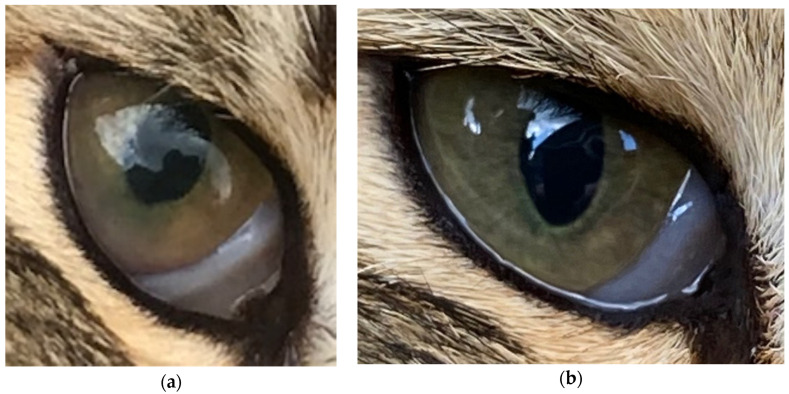
The cat’s eye before (**a**) and (**b**) after Mutian and topical steroid treatment. The cat presented with uveitis and keratic precipitates (**a**). On the right (**b**): the same eye after 7 days of systemic prednisolone, 2 weeks of topical prednisolone and 7 weeks of Mutian treatment, showing almost complete resolution of uveitis.

**Figure 2 viruses-12-01216-f002:**
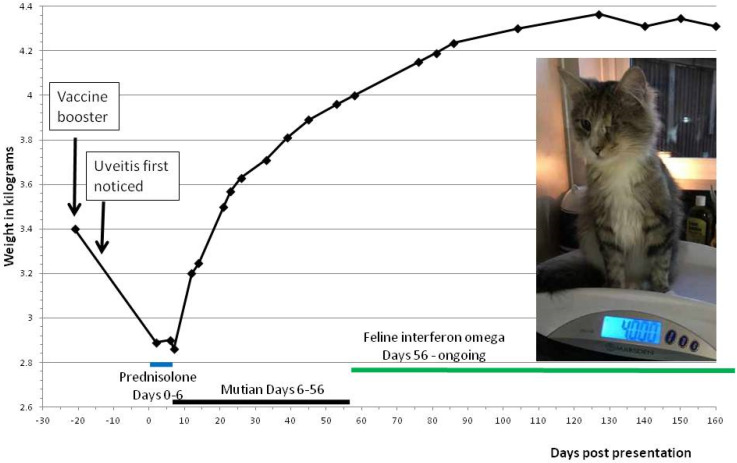
Weight before, during and after recovery from FIP and timeline. The cat’s weight is shown on the graph above: Day 0 represents the day of presentation at the clinic with uveitis. The vaccine booster was given 5 days prior to onset of uveitis, but the cat’s weight was not recorded prior to that day, so it is unknown if he was already losing weight when the booster was administered, or if weight loss occurred after the booster; i.e., if the FIP process was already underway when he was vaccinated, or if the vaccine triggered the onset of FIP. The cat lost 30 g of weight during systemic prednisolone treatment (Days 0 to 6), after which only topical steroids were given. Mutian treatment was given between Days 6 and 56; the cat’s weight rapidly increased following the onset of Mutian treatment, and weight gain continued after Mutian was replaced by feline interferon omega, until his weight leveled out at around 4.35 kg.

**Figure 3 viruses-12-01216-f003:**
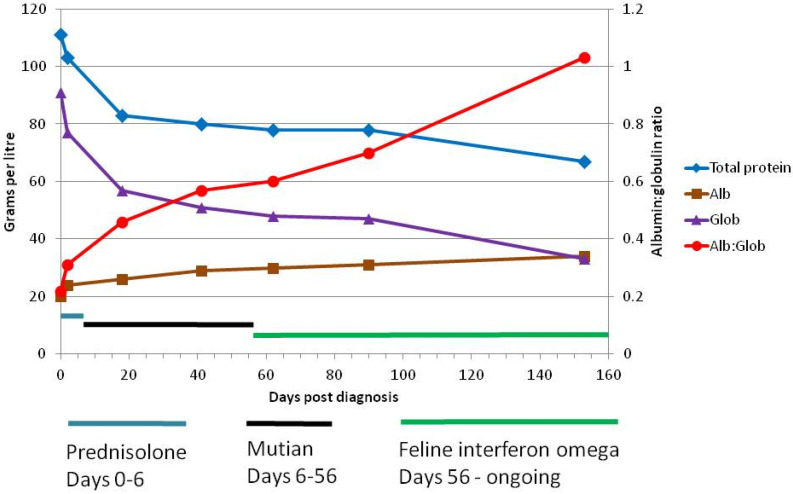
Treatment timeline in relation to albumin globulin levels showing reduction in globulin (and therefore total protein) levels, increase in albumin and albumin to globulin ratio. Improvement in these parameters continued after the cessation of Mutian treatment, and with feline interferon omega treatment.

**Table 1 viruses-12-01216-t001:** Feline coronavirus (FCoV) antibody titre and RT-qPCR C*_T_* results.

		Day −29	Day −19	Day 7	Day 41
Skywise	2yo Norwegian forest cat (NFC) FIP		>10,240	C*_T_* 30	Neg
Paddy	2yo NFC persistent pyrexia, lethargy	>10,240		C*_T_* 18	Neg
Oliver	8 yo Domestic shorthair cat		>10,240	C*_T_* 20	Neg
Link	1 yo NFC		>10,240	C*_T_* 20	Neg
Zelda	1 yo NFC		640	Neg	Neg

Yo—year old. C*_T_*—Cycle threshold of FCoV RT-qPCR test performed on a faecal sample. The days are relative to the first presumptive diagnosis of feline infectious peritonitis (FIP) in Skywise on Day 0 (18 April 2020); i.e., his FCoV antibody titre was tested 19 days before the FIP diagnosis. Mutian treatment began for Skywise on Day 6 and for the other FCoV-infected cats on Day 18 for 5 days.

**Table 2 viruses-12-01216-t002:** Laboratory data pertinent to FIP treatment monitoring.

	AGP	TP	Alb	Glob	A:G	Bilirubin	Hct	Lymph. Count	ALT	AP	GGT	SDMA	Creat	Urea
Ref. Range	<500 μg/mL	57–89 g/L	22–40 g/L	28–51 g/L	>0.8 *	0–15 μmol/L	30–52%	0.92–6.88 × 10^9^	12–130 U/L	14–111 U/L	0–4 U/L	0–14 μg/dL	71–212 μmol/L	5.7–12.9 mmol/L
**Day 0**	ND	111	20	91	0.22	11	25.7	0.95	64	ND	ND	ND	ND	ND
**Day 2**	3100	103	24	77	0.31	ND	22.0	2.07	ND	ND	ND	ND	ND	ND
**Day 18**	700	83	26	57	0.46	6	34.4	3.61	27	32	0	ND	88	9.6
**Day 41**	400	80	29	51	0.57	5	35.1	3.85	67	40	0	28	93	8.3
**Day 62**	400	78	30	48	0.60	5	35.2	2.64	61	40	0	19	106	6.5
**Day 90**	300	78	31	47	0.70	6	40.0	4.15	81	40	0	14	113	10.0
**Day 153**	400	67	34	33	1.03	1	35.2	4.49	81	46	2	ND **	98	7.8

* A:G of over 0.8 has a strong negative predictive value for FIP [8]. ** SDMA not repeated because urinalysis was normal. AGP—alpha-1 acid glycoprotein; TP—total protein; Alb—albumin; Glob—globulin; A:G—albumin to globulin ratio; Hct—haematocrit; Lymph—lymphocyte; ALT—alanine aminotransferase; AP—alkaline phosphatase; GGT—gamma glutamyl transferase; SDMA—symmetric dimethylarginine; Creat—creatinine; ND = not done.

**Table 3 viruses-12-01216-t003:** Timeline of treatment relative to resolution of uveitis.

Treatment			Day	Ophthalmoscopic Findings
**Systemic prednisolone**			0	Day of presentation to primary veterinary surgeon.
			1	Eye examined using a standard ophthalmoscope. Cornea unremarkable, anterior chamber slightly cloudy, but sedation would have been required for detailed examination. Cat not visual and was distressed.
			2	Blood samples and mesenteric lymph node fine-needle aspirates taken under sedation.
	**Topical prednisolone acetate 1% tid**		4	Keratic (protein) precipitates on lens present, partially obscuring fundus. Pupillary light reflex present but reduced.
		**Mutian at 8 mg/k for 19 days, then 6 mg/kg**	6	Cat’s guardian reported that she thought the cat could see.
			17	Cat’s guardian reported that the cat chased a bird.
			18	Uveitis resolving, anterior chamber clear. Keratic precipitates as before.
			41	Uveitis appeared largely resolved, keratic precipitates as before. Fundus seen this time, possible bulla seen, resolving chorioretinitis, but cat’s temperament precluded detailed examination. Excellent visual acuity.

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
