# Peer review of "Rapid Resolution of Non-Effusive Feline Infectious Peritonitis Uveitis with an Oral Adenosine Nucleoside Analogue and Feline Interferon Omega"

_viruses, 2020, doi:10.3390/v12111216_

Round 1

Reviewer 1 Report

This is a very interesting case report which offers a detailed description of a cat with feline infectious peritonitis (FIP) uveitis, that is positive for feline coronavirus (FCoV), and was successfully treated with the oral adenosine nucleotide analogue Mutian. This cat was very ill at presentation with systmic disease symptoms and an unfavorable prognosis. The clinical findings, treatment regiment and follow-up parameters are presented clearly. The significance of the case report in the context of current veterinary care is insightfully discussed. Specific comments:

1. Regarding the other household cats with FCoV, what were their general range of symptoms? Did any require specific treatments?

2. Regarding the uveitis: Was a slit lamp or comparable examination done or were additional clinical details of the uveitis available (such as whether posterior uveal tissues or the retina were involved, assessment of cells, etc.)? Was uveitis improved at all after oral prednisolone? What was the topical steroid drug and treatment regiment and duration? Could the authors comment on the potential efficacy of steroids versus Mutian in the resolution of the ocular inflammation? Is it known how common uveitis occurs in FCoV or FIP?

Figure 1: The image in (a) appears to show a mild synechiae, is this consistent with clinical observations? The first line of the figure legend should state "...and (b) after Mutian AND TOPICAL STEROID treatment...".

Author Response

First we would like to thank the reviewers most sincerely for their encouragement, for their thoroughness and the time they spent carefully reading our paper: you have added much value to the paper with your insightful comments.

Our replies are in italics.

Changes in the manuscript are highlighted in yellow.

Skywise Review 1

Open Review

Comments and Suggestions for Authors

This is a very interesting case report which offers a detailed description of a cat with feline infectious peritonitis (FIP) uveitis, that is positive for feline coronavirus (FCoV), and was successfully treated with the oral adenosine nucleotide analogue Mutian. This cat was very ill at presentation with systmic disease symptoms and an unfavorable prognosis. The clinical findings, treatment regiment and follow-up parameters are presented clearly. The significance of the case report in the context of current veterinary care is insightfully discussed.

Thank you very much indeed for your praise and encouragement.

Specific comments:

  1. Regarding the other household cats with FCoV, what were their general range of symptoms? Did any require specific treatments?

Paddy was presented with pyrexia on 18th March which led to the initial FCoV antibody titre testing of all cats, his blood sample revealed only a mild increase in globulins but no other indicators suggestive of FIP (e.g. lymphopenia, anaemia): he was treated with Convenia and meloxicam. The other FCoV-infected cats were asymptomatic but were treated with Mutian to prevent re-infection of Skywise.

  1. Regarding the uveitis: Was a slit lamp or comparable examination done or were additional clinical details of the uveitis available (such as whether posterior uveal tissues or the retina were involved, assessment of cells, etc.)?

Mark Fosbery replies: No slit lamp examination was done; the eye was examined using a standard ophthalmoscope.  Please see Table 3 below.

Was uveitis improved at all after oral prednisolone?

This was a fantastic question, but not an easy one to answer.  Therefore we returned to the patient’s medical records and made a timeline table with treatment and ophthalmologist’s comments, as shown below, which we have included in the paper.  By Day 4 post diagnosis, the patient had had 5 of the 6 days systemic corticosteroids which may have alleviated his pain somewhat, but had not made a big impact on the appearance of his uveitis.  However, his guardian reported some return of vision on Day 6. Unfortunately there was no veterinary examination between Days 4 and 18 (UK COVID lockdown restricted veterinary visits during this time).

Table 3. Timeline of treatment relative to resolution of uveitis

Treatment

Day

Ophthalmoscopic findings

Systemic prednisolone

0

Day of presentation to primary veterinary surgeon.

1

Eye examined using a standard ophthalmoscope. Cornea unremarkable, anterior chamber slightly cloudy, but sedation would have been required for detailed examination. Cat not visual and distressed.

2

Blood samples and mesenteric lymph node fine needle aspirates taken under sedation.

Topical prednisolone acetate 1% tid

4

Keratic (protein) precipitates on lens present, partially obscuring fundus.  Pupillary light reflex present but reduced.

Mutian at 8mg/k for 19 days, then 6mg/kg

6

Cat’s guardian reported that she thought the cat could see.

17

Cat’s guardian reported that the cat chased a bird.

18

Uveitis resolving, anterior chamber clear. Keratic precipitates as before.

41

Uveitis appeared largely resolved, keratic precipitates as before. Fundus seen this time, possible bulla seen, ?resolving chorioretinitis, but cat's temperament precluded detailed examination. Excellent visual acuity.

What was the topical steroid drug and treatment regiment and duration?

The topical steroid was prednisolone acetate treatment 1% (Pred forte), which was started on the 22nd April (Day 4), 1 drop to the right eye three times daily. This was continued for two weeks which we have now added to the paper on line 102.

Could the authors comment on the potential efficacy of steroids versus Mutian in the resolution of the ocular inflammation?

Based on Table 3 above, the following has been put into the Discussion (line 215):

We don’t know whether it was the Mutian, the systemic and topical steroids, or the combination of all treatments which effected the total resolution of the cat’s uveitis. The cat’s guardian reported some return of vision on Day 6 which was the day Mutian began, indicating that it was the steroids which were responsible for his return of sight (Table 3). Legendre et al (2017) [23] reported the progress of three of 17 cats with ocular manifestations of FIP treated with polyprenyl immunostimulant (PI). Their initial signs included anterior uveitis (all 3 cats), keratic precipitates (one cat), iris discoloration (one cat), and anisocoria (one cat). In two cats the anterior uveitis was significantly improved or resolved after 2 months on PI treatment with no corticosteroids. In the third cat the uveitis did not improve: the cat was receiving topical ocular corticosteroids concurrently with PI, and the eyes were enucleated [23]. Whether or not Mutian alone can effect a cure of FIP-related uveitis without concurrent topical corticosteroids remains to be seen. Systemic corticosteroids would appear to be of little or no value in these cases.

Is it known how common uveitis occurs in FCoV or FIP?

This is a good question: Legendre et al (2017) found uveitis in 17 of 59 (29%) cats with non-effusive FIP.  Norris et al (2005) found uveitis in 5 of 42 (12%) FIP cases (wet and dry not differentiated) and Bell et al (2006) from the Norris group reported FIP uveitis in 14 cats, but nowhere in the Bell paper does it state how many FIP cats were checked for uveitis.

Jinks et al (2016) from North Carolina Veterinary School asked the inverse of the question: i.e. how many cats with uveitis have FIP?  They published a study of 120 cats with uveitis: 19 cats, (15.8%) were diagnosed with FIP. 

I (DDA) have put the following summary of the above into Discussion:

The cat’s clinical signs of weight loss, poor appetite, diarrhoea, and uveitis were also congruent with a diagnosis of non-effusive FIP: FIP is a major cause of uveitis in young cats: ophthalmologists from North Carolina Veterinary School diagnosed FIP in 19 of 120 cats (15.8%) with uveitis [22].  Uveitis was reported in 17 of 59 (29%) cats with non-effusive FIP [23].  

Figure 1: The image in (a) appears to show a mild synechiae, is this consistent with clinical observations?

Mark Fosbery replies: No synechiae were seen.

The first line of the figure legend should state "...and (b) after Mutian AND TOPICAL STEROID treatment...".

Very good point, thank you: changed as requested.

 Diane says: I would just like to thank you very much indeed for your review: you have given us the questions that will most interest ophthalmologists who read our paper and which I – as a virologist – had not thought of: you exercised Mark’s little grey cells.  I think this was one of the best reviews I have ever had: our paper is very much better thanks to you.

Reviewer 2 Report

This study describes the resolution of non-effusive feline infectious peritonitis uveitis in a 2-year-old male neutered Norwegian Forest Cat with an oral adenosine nucleoside analogue and feline interferon omega, using the alpha-1 acid glycoprotein (AGP) as an indicator of recovery.

Broad comments: Overall, the article is very well written. Results are very interesting for FIP therapy, considering that until recently FIP has been considered an almost uniformly fatal disease.

A major comment concerns the absence of RT-qPCR among diagnostic tests to evaluate FIP treatment monitoring. In the opinion of this reviewer a sentence explaining why RT-qPCR was not used for treatment monitoring should be added in the discussion section.

Minor comments:

1. lines 48-49: in the opinion of this reviewer, the authors should better explain that the pyrexic in-contact cat was one of the two Norwegian Forest Cat introduced to the household two months previously (as reported by the authors in the discussion section, lines 155-156). Was this cat FIP-suspected because of diagnostic tests performed previously? This information may be useful because if no diagnosis was performed previously, the introduction into a household of kittens that were not tested for FCoV infection further confirms the typical risk factor for FIP cases (as also reported by the authors in the discussion section, lines 155-156).

2. lines 46-49: in the opinion of this reviewer it would be interesting to include information on when the blood sample was collected for RT-qPCR that resulted negative (as reported in the discussion section, in lines 138-139).

3. line 59: please include reference for M1058L and S1060A mutations. Moreover, in the opinion of this reviewer, it would be interesting to report the Ct value if it is indicative of a high viral load. Despite the lower sensitivity of real-time RT-PCR detecting S gene mutations compared to real-time RT-PCR detecting the FCoV 7b gene, a positive RT-qPCR result with a high viral load is very suggestive for FIP and would further confirm FIP diagnosis in the cat of this study.

4. lines 74-75: how was the Alpha-1 acid glycoprotein measured?

5. lines 85-86: was the pyrexic kitten among the three housemates that were also positive for FCoV RNA in their faeces? Did the kitten have a high viral load in faeces as defined by the Ct? Did the three cats have high viral loads in faeces?

6. line 105: were faeces also collected and tested by RT-qPCR at day 13th of Mutian treatment?

lines 120-122: the authors should include when (according to the timeline) the in-contact cats were treated with Mutian and when the faeces of all five cats were collected and confirmed negative.

line 219: Do the authors have information on the clinical status of the cat after the end of their study? In the opinion of this reviewer this information may further confirm the success in the cure of the cat with non-effusive FIP.

Author Response

First we would like to thank the reviewers most sincerely for their encouragement, for their thoroughness and the time they spent carefully reading our paper: you have added much value to the paper with your insightful comments.

Our replies are in italics.

Changes in the manuscript are highlighted in yellow.

This study describes the resolution of non-effusive feline infectious peritonitis uveitis in a 2-year-old male neutered Norwegian Forest Cat with an oral adenosine nucleoside analogue and feline interferon omega, using the alpha-1 acid glycoprotein (AGP) as an indicator of recovery.

Broad comments: Overall, the article is very well written. Results are very interesting for FIP therapy, considering that until recently FIP has been considered an almost uniformly fatal disease.

Many thanks for these kind and encouraging words.

A major comment concerns the absence of RT-qPCR among diagnostic tests to evaluate FIP treatment monitoring. In the opinion of this reviewer a sentence explaining why RT-qPCR was not used for treatment monitoring should be added in the discussion section.

Indeed we agree that RT-qPCR testing of faeces (but not blood) is a very important part of monitoring the response to FIP treatment, we have added a new Table 1 and expanded this sentence in Discussion:

We attribute our success to first, assessing that the anti-viral is effective against the FCoV strain infecting the cat by monitoring the quantity of virus shed in the faeces (by RT-qPCR) to ensure that viral load is reducing in response to the anti-viral; secondly preventing re-infection …

Minor comments:

  1. lines 48-49: in the opinion of this reviewer, the authors should better explain that the pyrexic in-contact cat was one of the two Norwegian Forest Cat introduced to the household two months previously (as reported by the authors in the discussion section, lines 155-156).

Actually the cat was one of the three cats already in the household: thank you for pointing out the ambiguity of our text, giving us the opportunity to clarify it. The cat’s guardian is happy for the cats’ names to be used in the new table and in the text where necessary in order to minimise confusion.  

Was this cat FIP-suspected because of diagnostic tests performed previously?

I’m not sure if this question refers to Skywise (the FIP case in this report) or Paddy, the first pyrexic cat? FIP was suspected in Skywise because he presented with fairly rapid onset uveitis and blindness within five days of his vaccine: at first it was thought to be a possible side effect (i.e. iatrogenic).  FIP was first suggested by his ophthalmologist, but suspicion of FIP was further encouraged by finding hyperglobulinaemia and low albumin to globulin ratio.

A viral condition was suspected in Paddy because of his lethargy and persistent pyrexia: FeLV and FIV were negative, but his FCoV antibody titre very high therefore the other cats were FCoV antibody tested to establish if Paddy would need to be separated from them or not (Table 1).

This information may be useful because if no diagnosis was performed previously, the introduction into a household of kittens that were not tested for FCoV infection further confirms the typical risk factor for FIP cases (as also reported by the authors in the discussion section, lines 155-156).

Thank you for this comment, in the www.catvirus.com diagnostic algorithm the first step in FIP diagnosis is the history, as you say – and is a very important step; then second step is the clinical examination. I have added a sentence to clarify that not only were the young cats a possible source of FCoV infection, but that their introduction was a stressor in addition to the vaccine.  We have also added sentences in Discussion about the prevalence of FIP in cats with uveitis:

The cat’s clinical signs of weight loss, poor appetite, diarrhoea, and uveitis were suggestive of non-effusive FIP: FIP is a major cause of uveitis in young cats: ophthalmologists from North Carolina Veterinary School diagnosed FIP in 19 of 120 cats (15.8%) with uveitis [22].  Uveitis was reported in 17 of 59 (29%) cats with non-effusive FIP [23]. 

  1. lines 46-49: in the opinion of this reviewer it would be interesting to include information on when the blood sample was collected for RT-qPCR that resulted negative (as reported in the discussion section, in lines 138-139).

The FNA and blood samples were taken on the 20th of April – i.e. Day 2 post presentation – and were sent together to Idexx, who—for reasons best known to themselves—did the RT-PCR test on the blood, not the MLN FNA as requested. Day 2 has now been added into what is now line 156.

  1. line 59: please include reference for M1058L and S1060A mutations.

The Chang reference has been added at this point.

Moreover, in the opinion of this reviewer, it would be interesting to report the Ct value if it is indicative of a high viral load. Despite the lower sensitivity of real-time RT-PCR detecting S gene mutations compared to real-time RT-PCR detecting the FCoV 7b gene, a positive RT-qPCR result with a high viral load is very suggestive for FIP and would further confirm FIP diagnosis in the cat of this study.

I totally agree with you that the Ct value would have been very helpful but unfortunately Idexx laboratory does not report Ct, despite repeated requests from myself (DDA) to them to do so. For this reason I requested that the faecal samples be sent to VDS instead of Idexx, so that a Ct would be reported.

  1. lines 74-75: how was the Alpha-1 acid glycoprotein measured?

AGP was measured at Idexx Laboratory and this has been added to the manuscript: unfortunately they don’t disclose their exact method although I suspect it may have been outsourced to Glasgow University’s Veterinary Diagnostic Services.

  1. lines 85-86: was the pyrexic kitten among the three housemates that were also positive for FCoV RNA in their faeces? Did the kitten have a high viral load in faeces as defined by the Ct? Did the three cats have high viral loads in faeces?

In response to this reviewer’s interest in the results of the in contact cats we have added a table of their results (Table 1).  Your question is really quite uncanny: you are quite correct - the pyrexic cat (actually a two year old) had the highest virus load, as indicated by the lowest faecal Ct.  Skywise had the lowest viral load, but had had one, perhaps two, doses of Mutian when the sample was taken.

  1. line 105: were faeces also collected and tested by RT-qPCR at day 13th of Mutian treatment?

The faecal test dates and results are now listed in Table 1, showing that Mutian did clear FCoV from the faeces of Skywise and his house-mates.

lines 120-122: the authors should include when (according to the timeline) the in-contact cats were treated with Mutian and when the faeces of all five cats were collected and confirmed negative.

As stated above, a new table has been added to the paper and we thank you for allowing us the opportunity to expand on this aspect of the case report.

line 219: Do the authors have information on the clinical status of the cat after the end of their study? In the opinion of this reviewer this information may further confirm the success in the cure of the cat with non-effusive FIP.

We have updated Table 2 and Figures 2 and 3 with the most recent weight and blood results to show that improvement in the cat’s condition has continued and that he is alive and well now 5 months post-diagnosis.  We will be following him over the coming years.

 Thank you for your review which has so greatly improved our paper. Your review allowed for expansion of what happened to the in-contact cats and I was so grateful to find in you somebody who really “gets” the importance of Cts in FCoV RT-PCR testing.